# To update or not to update?
# Neurons at equilibrium in deep models

**Andrea Bragagnolo**
University of Turin, Italy
Synesthesia s.r.l., Turin, Italy
andrea.bragagnolo@unito.it

**Enzo Tartaglione**
LTCI, Télécom Paris,
Institut Polytechnique de Paris, France
enzo.tartaglione@telecom-paris.fr

**Marco Grangetto**
University of Turin, Italy
marco.grangetto@unito.it

## Abstract

Recent advances in deep learning optimization showed that, with some a-posteriori information on fully-trained models, it is possible to match the same performance by simply training a subset of their parameters. Such a discovery has a broad impact from theory to applications, driving the research towards methods to identify the minimum subset of parameters to train without look-ahead information exploitation. However, the methods proposed do not match the state-of-the-art performance and rely on unstructured sparsely connected models.

In this work we shift our focus from the single parameters to the behavior of the whole neuron, exploiting the concept of neuronal equilibrium (NEq). When a neuron is in a configuration at equilibrium (meaning that it has learned a specific input-output relationship), we can halt its update; on the contrary, when a neuron is at non-equilibrium, we let its state evolve towards an equilibrium state, updating its parameters. The proposed approach has been tested on different state-of-the-art learning strategies and tasks, validating NEq and observing that the neuronal equilibrium depends on the specific learning setup.

## 1 Introduction

In recent years, deep learning has become a staple solution to different tasks, such as computer vision, bio-informatics, speech recognition, and many more. Unfortunately, modern neural network architectures are becoming more and more "expensive": performing simple inferences requires a lot of computational resources, and training the models even more. Such cost poses several challenges for the research community: the training of a network model is associated with large carbon footprints and the commercialization of AI research (especially for edge devices) is hindered by the resource requirements of the models Strubell et al. (2019).

For several years now, many works in literature have shown that is possible to shrink both the size and resource requirements, mainly via quantization Yang et al. (2020); Jin et al. (2022) and pruning Wang et al. (2020); Tang et al. (2020); Tanaka et al. (2020). In particular, pruning techniques can drastically reduce the number of operations needed to perform inference, without affecting the overall performance of the model. Pruning targets the reduction of parameters in the model, which has advantages after training; unfortunately, it is unable to reduce the cost of the training cycle, and on the contrary, it requires, in general, iterations of few-shot pruning, followed by fine-tuning Lee et al. (2019); Tartaglione et al. (2018).

This paper has been accepted for publication at the 36th Conference on Neural Information Processing Systems (NeurIPS 2022).

A recent work shows the existence of sub-graphs in the whole deep learning model which, when trained in isolation, match the performance of the whole model Frankle and Carbin (2019). This opens the road towards a whole field of research, where many approaches are proposed to find these parameters a-priori: this shows a potential environmental impact, considering that a smaller model is trained, leading to a lower energy consumption Strubell et al. (2019). Indeed, when training a neural network, the back-propagation procedure and the weights update lead to the larger part of FLOPs (compared to forward propagation) Plaut et al. (1986); Baydin et al. (2018). Researchers started to experiment pruning early in the training or even before the training starts in the hope of training a restricted number of parameters. Unfortunately, training a sparse network with standard optimizers leads to subpar results Evci et al. (2019) or the final result does not differ much from magnitude pruning at the end of the training Frankle et al. (2021). The causes for such behaviors are still a matter of debate among the community 2019; 2021.

In this work, we shift the focus from the single parameter to the whole neuron, and we propose NEq, an approach to evaluate whether a given neuron is at *equilibrium* for the learning dynamics. If the neuron is in such a state, its parameters have already reached a target configuration and do not require a further update. Unlike many other recent approaches, NEq disables *entire neurons* (hence, in a structured way), does not require prior knowledge of the specific task (for example by first training a model to convergence), and automatically self-adapts to the specific learning policy deployed. Unlike pruning techniques, NEq does not remove the neurons' contribution to the output; instead, it only prevents unnecessary updates to their weights: as a result, we reduce the number of operations performed by the back-propagation algorithm and the optimizer.

The rest of the paper is organized as follows. In Sec. 2 we discuss the related literature; Sec. 3 presents the concept of neuronal equilibrium and how to evaluate it; Sec. 4 presents the experimental validation inclusive of an ablation study and Sec. 5 draws the conclusions.

## 2 Related works

It is broadly acknowledged that the typically-deployed deep learning models on the state-of-the-art scenarios are over-parametrized Mhaskar and Poggio (2016); Brutzkus et al. (2018). This ignites two lines of research: reducing the size of these models (with *pruning* algorithms) or saving computational resources at training time. While the first has been broadly explored, the latter suffered a stalemate until a recent work suggested its feasibility Frankle and Carbin (2019).

**Pruning strategies.** Attempts to reduce the number of parameters from learned models date back to 1989 when Mozer and Smolensky proposed *skeletonization*, a technique to identify less relevant neurons in a trained model and to remove them 1989. This was accomplished thanks to the evaluation of an error function penalty from a pre-trained model. In the same years, LeCun et al. also proposed a work where the information from the second-order derivative of the error function is leveraged to rank the parameters of the trained model on a saliency-like basis LeCun et al. (1990). In the last decade, thanks to the broad availability of computational resources, pruning approaches gained-back popularity, with approaches including the exploitation of dropout Molchanov et al. (2017); Gomez et al. (2019), sensitivity-based approaches Lee et al. (2018); Tartaglione et al. (2021, 2022), relaxation of $\ell_0$ regularization Louizos et al. (2017); Sanh et al. (2020) and optimization of auxiliary parameters Xiao et al. (2019). Despite leading to very compact deep models, the demanded training complexity is frequently very high, as many of the most well-known approaches rely on iterative fine-tuning (or few-shot pruning) approaches. Hence, the training complexity for these techniques is larger than training a vanilla model (which in many cases is used as initialization). Much computational complexity could be saved if structured pruning is applied before training the model itself (namely, entire neurons/channels are pruned). Some works on pruning have suggested the possibility of re-allocating previously pruned parameters Tresp et al. (1996) or filters He et al. (2018) to learn new functions and minimize the loss, towards higher generalization capability Blalock et al. (2020).

**The lottery ticket hypothesis.** In a recent paper, Frankle and Carbin provided empirical evidence that, at initialization, there exists a sub-graph of the original deep learning model such that, when trained in isolation, it can match the performance of the complete model Frankle and Carbin (2019). Indeed, the authors claim that the parameters in the sub-graph have "won at the lottery of initialization". From a very practical perspective, this means that all the parameters not in the selected sub-graph are, de facto, pruned from the model, not requiring any gradient computation. This empirical evidence potentially opens the road to the development of algorithms for pruning at initialization. Unfortunately, at such point, two obstacles are yet to be tackled. From one side, Frankle and Carbin (2019) uses an iterative

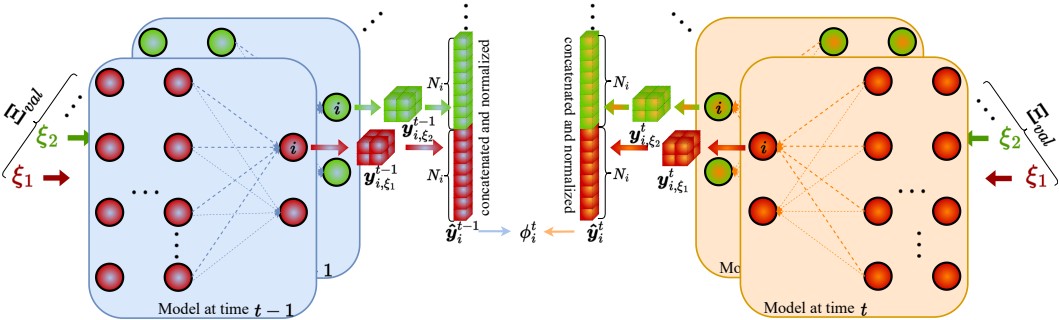

Figure 1: For a given time $t$ the model (either in blue or orange) receives samples from the validation set (in red or green). The output of the $i$-th neuron (whose cardinality is $N_i$) depends on both the model's parameters and the specific sample on the validation set. These outputs are squeezed, concatenated and the obtained vector (of size $N_i \cdot \|\Xi_{val}\|_0$, being $\|\Xi_{val}\|_0$ the cardinality of the validation set) is then normalized, obtaining $\hat{\boldsymbol{y}}_i^t$.

procedure to identify the sub-graph: it simply proves its existence but does not provide a method to find it directly at initialization. On another side, it focuses on un-structured sub-graphs, meaning that parameters are treated in isolation: is it possible to find structured sub-graphs (or in other words, removing entire nodes and not just arcs)?

**Beyond the lottery ticket hypothesis.** Many works of the last two years have received inspiration from the lottery ticket hypothesis, in the quest of determining the early lottery tickets. This is, however, a tough task to solve: Frankle et al. (2020) shows that there is a region, at the very early stages of learning, where the lottery tickets identified with iterative pruning are "not stable" (meaning that tickets extracted at different moments of this early stages are essentially different). This suggests that, in the very first epochs, the neural network evolves in very different states, making the problem of a-priori identifying winning tickets hard Tartaglione (2022). This is also endorsed by other works, including Morcos et al. (2019); Malach et al. (2020). On the other hand, other approaches reduce the overall complexity of the iterative training by drawing early-bird tickets You et al. (2019) (meaning that they learn the lottery tickets when the model has not yet reached full convergence), even reducing the training data Zhang et al. (2021) or moving the first steps towards structurally-sparse winning tickets, yet still at iterative fashion, applying similar concept as Frankle and Carbin to entire neurons and channels Chen et al. (2022).

With NEq we learn the important lessons related to the lottery ticket hypothesis, targeting the reduction of computational complexity at training time, without exploiting knowledge on pre-training models, or rewinding. Hence, we do not target the achievement of sparse architectures, but we aim at determining when a whole neuron requires to be updated or when the computation of the gradient for its parameters is not necessary. As such, the comparison with the other presented approaches will be in general unfair, as they require a much greater computational complexity for training because of un-structured sparsity (which introduces an overhead in the representation of the tensors) and the iterative strategies. Furthermore, we will observe the possibility, along the training process, that some neurons, already kept in a "frozen" state, might unfreeze, requiring additional update steps. Although resource re-allocation has been exploited before Tresp et al. (1996); He et al. (2018), our unfreezing is different as it involves learning of a specific target function by the neuron, and not learning new ones through their re-allocation. In the next section, the notion of neuronal equilibrium will be presented, as well as the strategy to determine which neurons will require gradient computation.

## 3 Neurons at equilibrium

In this section, we will treat the problem of determining when a given neuron, along with the learning dynamics, finds itself at equilibrium. Towards this end, we define $\boldsymbol{y}_{i,\xi}^t$ as the output of the $i$-th neuron when the input $\xi$ is fed to the whole model trained after $t$ epochs. Given a set of inputs $\xi \in \Xi_{val}$ (where $\Xi_{val}$ is the validation set), it is possible to compare each $n$-th element $y_{i,n,\xi}^t$ with $y_{i,n,\xi}^{t-1}$, for the same model's input: what changes are the parameters of the model. Fig. 1 provides an overview of the nomenclature used: in the rest of the section we will see how to determine when a neuron is at equilibrium.

## 3.1 Neuronal equilibrium

In this section, we are interested in evaluating when the relationship between the input of the model and the output of the $i$-th neuron is modified. When this happens, the neuron is at *non-equilibrium*, meaning that its learned function, in the whole picture (or in other words, taking into account the evolution of the neurons in the previous layers as well), is still "evolving". We are interested in identifying the scenarios where the neuron is at *equilibrium* at the net of the interactions with the other neurons. To assess it, let us define the cosine similarity between all the outputs of the $i$-th neuron at time $t$ and at time $t-1$ for the whole validation set $\Xi_{val}$ as

$$\phi_i^t = \sum_{\xi \in \Xi_{val}} \sum_{n=1}^{N_i} \hat{y}_{i,n,\xi}^t \cdot \hat{y}_{i,n,\xi}^{t-1}. \tag{1}$$

Here we can determine that, when $\phi_i = 1$, the $i$-th neuron produces the same (eventually scaled) output between the evaluation at time $t$ and at time $t-1$ for the same input $\xi$ of the model. We say the $i$-th neuron reaches equilibrium when we have

$$\lim_{t \to \infty} \phi_i^t = k, \tag{2}$$

where $k \in [-1; +1]$ is some constant value. We can have the following scenarios:

- $k = 1$. In this case, the two outputs are perfectly correlated, meaning that the relationship bounding the input of the whole model $\xi$ and the output of the specific $i$-th neuron is maintained.

- $k \in (0; 1)$. The outputs correlate, but we are in presence of an oscillatory behavior (in the sense that the cosine similarity varies by a constant value between consecutive evaluations). This effect can be caused by stochastic effects like high learning rate/regularization, small batch size, or a combination of them.

- $k \in [-1; 0]$. Also in this case we are in the presence of oscillatory behavior, but the outputs are anti-correlated or de-correlated.

Here follows the evaluation framework to determine the arrival to an equilibrium state for the $i$-th neuron.

## 3.2 Neuron dynamics evaluation

To assess the convergence to equilibrium for (2), it is important to evaluate the variation of the similarities $\phi_i^t$ over time. Towards this end, let us introduce the *variation of similarities*

$$\Delta\phi_i^t = \phi_i^t - \phi_i^{t-1}. \tag{3}$$

According to the analysis in Sec. 3.1, in this case, we say we reach equilibrium when $\Delta\phi_i^t \to 0$. Hence, it is useful to keep track of the recent evolution over the similarity scores in the model: towards this end, we can introduce the velocity of the similarity variations:

$$v_{\Delta\phi_i}^t = \Delta\phi_i^t - \mu_{eq} v_{\Delta\phi_i}^{t-1}, \tag{4}$$

where $\mu_{eq}$ is the momentum coefficient. We can rewrite (4) making the similarity scores explicit, obtaining

$$v_{\Delta\phi_i}^t = \begin{cases} \phi_i^t + \sum_{m=1}^{t} (-1)^m \left[ (\mu_{eq})^{m-1} + (\mu_{eq})^m \right] \phi_i^{t-m} & \mu_{eq} \neq 0 \\ \phi_i^t - \phi_i^{t-1} & \mu_{eq} = 0, \end{cases} \tag{5}$$

where $(\cdot)^m$ indicates power of $m$. If we assume $\phi^t \in [0; 1] \forall t$ (which is the case of ReLU-activated neurons), in order to prevent (5) from exploding, we need to set $\mu_{eq} \in [0; 0.5]$. We can extend this setup to any layer if we assume that the neurons in the trained model will not reach equilibrium with anti-correlated outputs.

### 3.3 Selection of trainable neurons at non-equilibrium

To evaluate when a given neuron has reached equilibrium, exploiting (2), we can say that the $i$-th neuron is at equilibrium when it can satisfy

$$\left| v_{\Delta\phi}^t \right| < \varepsilon, \quad \varepsilon \geq 0. \tag{6}$$

It is important to notice that, once (6) is satisfied for a certain $t$, in case something changes in the learning dynamics (for example, the learning rate is re-scaled), there may exist some $t' > t$ such that the constraint is not satisfied anymore. When this happens, it means that the neuron is driven towards *new states*, and is no anymore at equilibrium. Hence, it requires to be updated again.

### 3.4 Overall training scheme

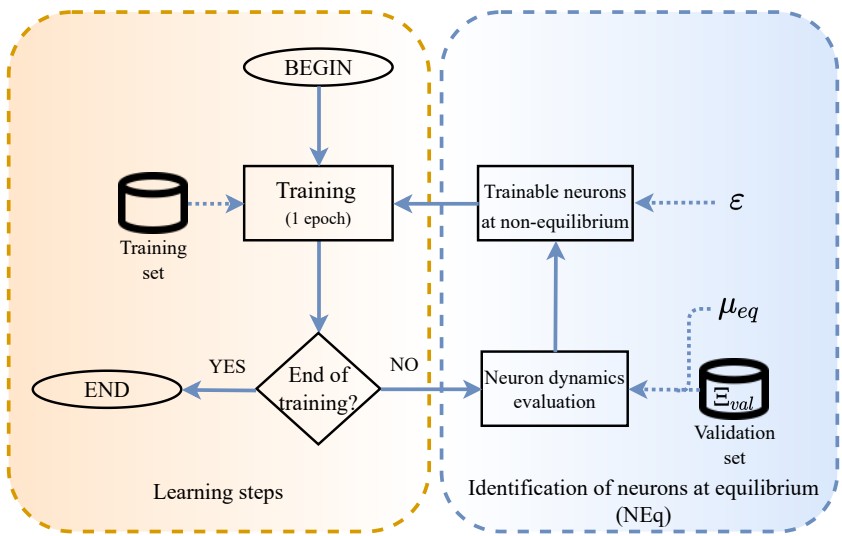

Figure 2: Overall training scheme. In orange is the standard training part and in blue is the neuron equilibrium evaluation and selection stages (we name this whole part NEq).

The overall training scheme is summarized in Fig. 2. The model is trained for one epoch, after which neurons at equilibrium are identified. We split this into two phases: in the first (neuron dynamics evaluation), the velocity of the similarities is evaluated according to (4), while in the second (trainable neurons at non-equilibrium) the hidden neurons at non-equilibrium, which will be trained for the next epoch, are identified according to (6). For the first epoch, all the neurons are considered at non-equilibrium by default. The evaluation of neurons at equilibrium is agnostic to the general training strategy, which can include arbitrary re-scaling for the learning rate/hyper-parameters or the most common optimizers. In the next section, we will test this procedure on very different architectures, tasks, and learning strategies.

## 4 Experiments

In this section, we report the experiments supporting the approach as presented in Sec. 3.4. First, we will perform an ablation study, analyzing single contributions for the introduced hyper-parameters and providing an overview of neuronal equilibrium along the training process; then, we will test the proposed technique on state-of-the-art network architectures, datasets, and learning policies. All experiments were performed using 8 NVIDIA A40 GPUs and the source code uses PyTorch 1.10.[1]

### 4.1 Ablation study

We performed our ablation study training a ResNet-32 He et al. (2015) model on CIFAR-10 Krizhevsky et al.. Unless differently specified, following the hyper-parameters setup of Zagoruyko

---

[1]the source code is available at `https://github.com/EIDOSLAB/NEq`.

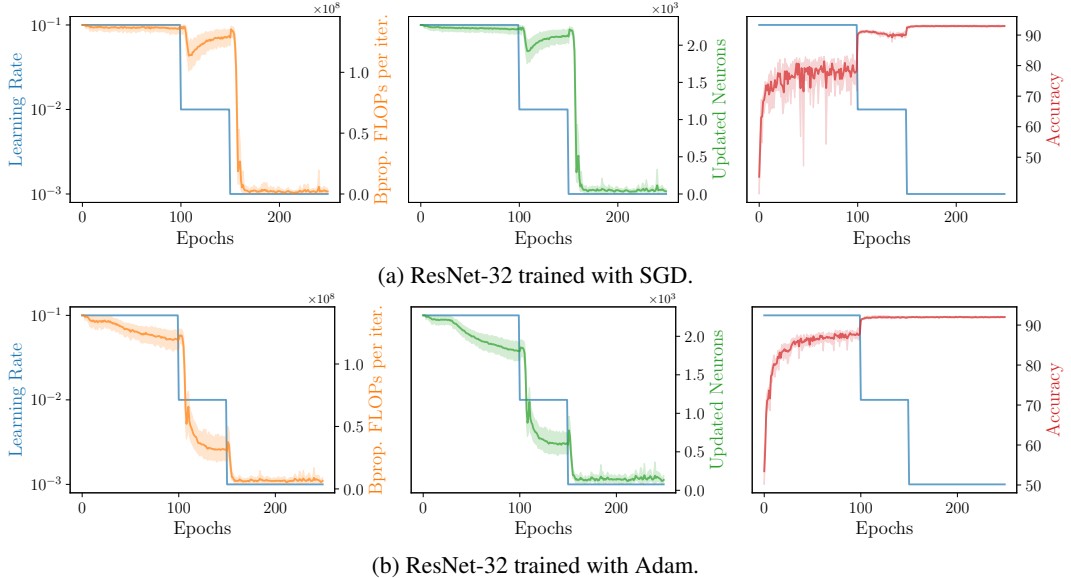

(a) ResNet-32 trained with SGD.

(b) ResNet-32 trained with Adam.

Figure 3: Back-propagation FLOPs (left, orange), updated neurons (center, green), and classification accuracy (right, red) for ResNet-32 trained on CIFAR-10.

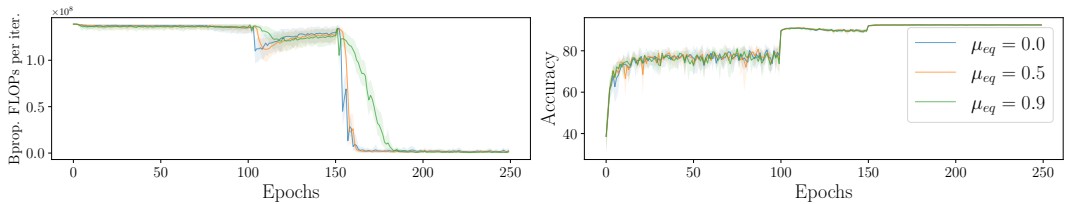

Figure 4: Back-propagation FLOPs (left) and accuracy (right) for different values of $\mu_{eq}$ for ResNet-32 trained on CIFAR-10.

and Komodakis (2016), the model is trained using SGD as an optimizer, with a starting learning rate $\eta = 0.1$ and momentum $\mu_{opt} = 0.9$ and a weight decay of $5 \times 10^{-4}$ for 250 epochs. The learning rate is decayed by a factor of 10 after 100 and 150 epochs, using the formulation as in (4), with $\|\Xi_{val}\|_0 = 50$, $\mu_{eq} = 0.5$, and $\varepsilon = 0.001$.

### 4.1.1 SGD vs Adam

To show that our technique automatically self-adapts to the training policy, we compare the evolution of the FLOPs required for a back-propagation step and the number of updated neurons of two different training of the ResNet-32: one using the SGD optimizer with $\mu_{opt} = 0.9$, and the other using the Adam optimizer. For Adam we leave the hyper-parameters to their default values ($\eta = 0.001$, $\beta_1 = 0.9$, and $\beta_2 = 0.999$) and use the same weight decay as for SGD ($5 \times 10^{-4}$). Fig. 3 shows the trends for the two training procedures. We can see that in the first phase of the train, where $\eta$ is high, the amount of the trained neurons (and the FLOPs required for the backward pass) is higher. This is related to the general lack of equilibrium in the neurons of the network: at high learning rates, the configuration of the neurons' parameters is subjected to high stochastic noise. As the train progresses, and the network progresses toward its final configuration, fewer and fewer neurons need to be updated. Noticeably, Adam drives the neurons towards equilibrium in a faster way, as expected; however, in simple tasks like the considered one, converges to lower accuracy scores (92.96% for SGD and 92.01% for Adam). Furthermore, at the first learning rate decay (epoch 100), for SGD the number of updated neurons first decreases and then increases, such a phenomenon is not present in the Adam case. This is explained by the different working principles of the two optimizers: SGD explores the solution space looking for large minima, searching for configurations that prevent equilibrium in high learning rate regimes.

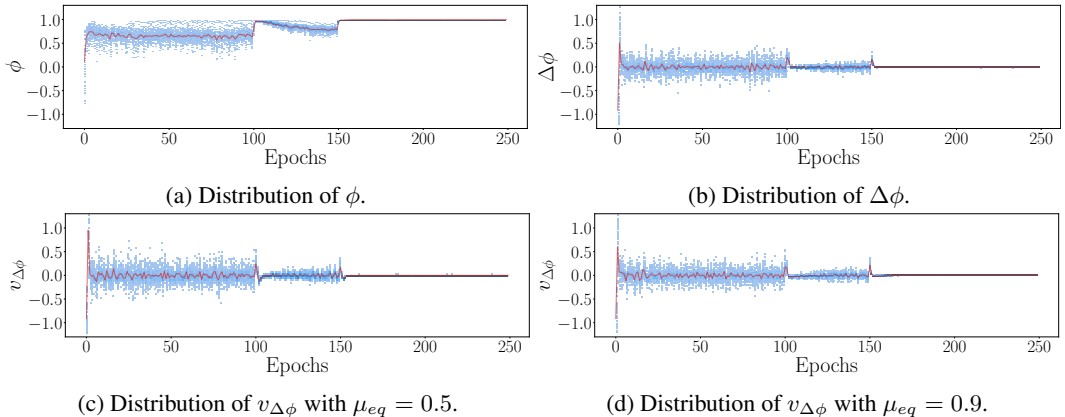

(a) Distribution of $\phi$.

(b) Distribution of $\Delta\phi$.

(c) Distribution of $v_{\Delta\phi}$ with $\mu_{eq} = 0.5$.

(d) Distribution of $v_{\Delta\phi}$ with $\mu_{eq} = 0.9$.

Figure 5: Neuronal equilibrium-related quantities for ResNet-32 trained on CIFAR-10. The red line indicates the average.

Table 1: Ablation for ResNet-32 trained on CIFAR-10.

(a) Ablation on $\|\Xi_{val}\|_0$.

| $\|\Xi_{val}\|_0$ | Bprop. FLOPs per iteration | Accuracy |
|---|---|---|
| 500 | 84.73M $\pm$ 629.15K | 92.70 $\pm$ 0.12 |
| 250 | 82.62M $\pm$ 613.14K | 92.80 $\pm$ 0.43 |
| 100 | 84.82M $\pm$ 628.05K | 92.81 $\pm$ 0.15 |
| 50 | 84.81M $\pm$ 629.12K | 92.96 $\pm$ 0.21 |
| 25 | 84.49M $\pm$ 629.37K | 92.62 $\pm$ 0.28 |
| 10 | 84.91M $\pm$ 627.11K | 92.70 $\pm$ 0.27 |
| 5 | 84.34M $\pm$ 619.37K | 92.57 $\pm$ 0.48 |
| 2 | 85.76M $\pm$ 617.11K | 92.80 $\pm$ 0.24 |
| 1 | 85.56M $\pm$ 626.09K | 92.77 $\pm$ 0.23 |

(b) Ablation on $\varepsilon$.

| $\varepsilon$ | Bprop. FLOPs per iteration | Accuracy |
|---|---|---|
| 0.0 | 136.71M $\pm$ 15.34K | 92.62 $\pm$ 0.23 |
| 0.0001 | 124.45M $\pm$ 161.76K | 92.65 $\pm$ 0.40 |
| 0.0005 | 89.29M $\pm$ 589.71K | 92.69 $\pm$ 0.19 |
| 0.001 | 83.62M $\pm$ 629.22K | 92.96 $\pm$ 0.21 |
| 0.005 | 65.65M $\pm$ 591.07K | 91.72 $\pm$ 0.37 |
| 0.01 | 52.53M $\pm$ 590.30K | 91.23 $\pm$ 0.32 |
| 0.05 | 16.10M $\pm$ 254.94K | 86.80 $\pm$ 0.29 |
| 0.1 | 4.97M $\pm$ 186.54K | 83.90 $\pm$ 0.66 |
| 0.5 | 1.78M $\pm$ 137.92K | 76.78 $\pm$ 2.57 |

### 4.1.2 Distribution of $\phi$ & choice of $\mu_{eq}$

Looking at different values for $\mu_{eq}$ in Fig. 4, we observe for all the values a convergence to similar accuracy. Despite without warranty from the theory, we tested a very large value for the momentum coefficient (0.9): the convergence of $v^t_{\Delta\phi_i}$ shows that the neurons are in general in a very correlated case of equilibrium, with very high values for $k$ in (2), which is also empirically observed in Fig. 5a. However, including a very large value for $\mu_{eq}$ maintains the memory of very old variations, producing a sub-optimal reduction in terms of FLOPs reduction. We find that a good compromise, supported by the findings as in Sec. 3.2, is to set $\mu_{eq}$ to 0.5. Fig. 5 reports the distribution for the velocities for $\phi$, $\Delta\phi$, and $v_{\Delta\phi}$, observing that the average converges to a specific $k$ for each of the three learning rates used.

### 4.1.3 Impact of the validation set size and $\varepsilon$

Tab. 1b provides an empirical evaluation of the impact on the performance and on the FLOPs varying the validation set size. We indeed observe not a significant impact on the performance of the model varying it. Interestingly, the approach produces extremely good results even for extremely low cardinality for the validation set (down to even a single image): this can be explained by the presence of convolutional layers (the only fully-connected layer is the output layer, excluded by default) which even with little images produce high-dimensionality output in every neuron (Fig. 1) and by the homogeneity of the considered dataset. Investigating the impact of $\varepsilon$, instead, we find for very high values of $\varepsilon$ a drop in performance, identifying a good compromise for classification tasks to 0.001.

Table 2: Results of the application of NEq to each experimental setup, compared to the stochastic approach. We report the average FLOPs per iteration at backpropagation and the final performance of the model evaluated on the test set (values annotated with $^\dagger$ report the classification accuracy, values annotated with $^\ddagger$ report the mean IoU).

| Dataset | Model | Approach | Bprop. FLOPs per iteration | Performance |
|---|---|---|---|---|
| CIFAR-10 | ResNet-32 | Baseline | 138.94M $\pm$ 0.0M | 92.85% $\pm$ 0.23%$^\dagger$ |
| | | Stochastic ($p = 0.2$) | 112.99M $\pm$ 0.00M (-18.68%) | 92.78% $\pm$ 0.19% (-0.07%)$^\dagger$ |
| | | Stochastic ($p = 0.5$) | 69.75M $\pm$ 0.00M (-49.8%) | 91.88% $\pm$ 0.27% (-0.97%)$^\dagger$ |
| | | Stochastic* | 86.34M $\pm$ 0.00M (-37.85%) | 92.23% $\pm$ 0.25% (-0.62%)$^\dagger$ |
| | | Neq | 84.81M $\pm$ 0.63M (-38.96%) | 92.96% $\pm$ 0.21% (+0.11%)$^\dagger$ |
| ImageNet-1K | ResNet-18 | Baseline | 3.64G $\pm$ 0.0G | 69.90% $\pm$ 0.04%$^\dagger$ |
| | | Stochastic ($p = 0.2$) | 2.94G $\pm$ 0.00G (-19.26%) | 69.42% $\pm$ 0.16% (-0.48%)$^\dagger$ |
| | | Stochastic ($p = 0.5$) | 1.85G $\pm$ 0.00G (-49.11%) | 69.18% $\pm$ 0.03% (-0.72%)$^\dagger$ |
| | | Stochastic* | 2.82G $\pm$ 0.00G (-22.58%) | 69.45% $\pm$ 0.06% (-0.45%)$^\dagger$ |
| | | Neq | 2.80G $\pm$ 0.03G (-23.08%) | 69.62% $\pm$ 0.06% (-0.28%)$^\dagger$ |
| | Swin-B | Baseline | 30.28G $\pm$ 0.00G | 84.71% $\pm$ 0.04% $^\dagger$ |
| | | Stochastic ($p = 0.2$) | 24.65G $\pm$ 0.00G (-18.6%) | 84.54% $\pm$ 0.04% (-0.83%)$^\dagger$ |
| | | Stochastic ($p = 0.5$) | 16.15G $\pm$ 0.00G (-46.67%) | 84.40% $\pm$ 0.02% (-0.31%)$^\dagger$ |
| | | Stochastic* | 11.02G $\pm$ 0.00G (-63.67%) | 84.27% $\pm$ 0.04% (-0.44%)$^\dagger$ |
| | | Neq | 10.78G $\pm$ 0.02G (-64.39%) | 84.35%$\pm$0.02% (-0.36%)$^\dagger$ |
| COCO | DeepLabv3 | Baseline | 305.06G $\pm$ 0.0G | 67.71% $\pm$ 0.02%$^\ddagger$ |
| | | Stochastic ($p = 0.2$) | 248.69G $\pm$ 0.00G (-18.48%) | 67.11% $\pm$ 0.02% (-0.60%)$^\ddagger$ |
| | | Stochastic ($p = 0.5$) | 163.42G $\pm$ 0.00G (-46.43%) | 66.91% $\pm$ 0.04% (-0.80%)$^\ddagger$ |
| | | Stochastic* | 229.00G $\pm$ 0.00G (-24.93%) | 67.02% $\pm$ 0.03% (-0.69%)$^\ddagger$ |
| | | Neq | 217.29G $\pm$ 0.04G (-28.77%) | 67.22% $\pm$ 0.04% (-0.49%)$^\ddagger$ |

## 4.2 Main experiments

In this section, we show the results of the proposed method. For our experiments, we focused on different state-of-the-art architectures trained on standard classification and semantic segmentation datasets. All the learning policies used are borrowed from other works and are un-optimized to test the adaptability of NEq.

**ResNet-32 trained on CIFAR-10**. The training spans 250 epochs, using SGD as optimizer with momentum $\mu_{opt} = 0.9$, weight decay $5 \times 10^{-4}$ and initial learning rate $\eta = 0.1$, reduced by a factor of 10 after 100 and 150 epochs. To evaluate the neuronal equilibrium we used a $\|\Xi_{val}\|_0$ of 50 images, $\mu_{eq} = 0.5$, and $\varepsilon = 0.001$. We used a batch size of 100 images during training.

**ResNet-18 trained on ImageNet-1K Krizhevsky et al. (2012)**. This model was trained with SGD as optimizer for 90 epochs, with $\eta = 0.1$, reduced by a factor of 10 every 30 epochs, $\mu_{opt} = 0.9$ and weight decay $10^{-4}$ using a batch size of 128. We used a $\|\Xi_{val}\|_0$ of 1.2k images, $\mu_{eq} = 0.5$, and $\varepsilon = 0.001$.

**Swin Transformer Liu et al. (2021) (Swin-B) trained on ImageNet-1K**. To test our technique on more modern models and training policies, we used the Swin-B architecture. Here we trained the model starting from a pre-trained checkpoint trained on ImageNet-21K, following the official GitHub repository[2] released under the MIT License. We used a $\|\Xi_{val}\|_0$ of 1.2k images, $\mu_{eq} = 0.5$, and $\varepsilon = 0.001$.

**DeepLabv3 Chen et al. (2017) trained on COCO Lin et al. (2014)**. Other than classification tasks of varying complexity, we tested our procedure on a semantic segmentation problem. For this experiment, we used DeepLabv3 with a ResNet-50 backbone and the COCO dataset. To train the

---

[2]`https://github.com/microsoft/Swin-Transformer`

network, we followed the state-of-the-art procedure defined in PyTorch[3]. We evaluated the neuronal equilibrium using a $\|\Xi_{val}\|_0$ of 320 images, $\mu_{eq} = 0.5$, and $\varepsilon = 0.02$.

### 4.3 Discussion

The results (average over five different runs) are reported in Tab. 2. For each experiment, we compare our technique with a "stochastic" approach. Namely, we randomly halt, at every epoch and with probability $p$, the update of a given neuron. We test on three different probabilities: $0.2$, $0.5$, and a probability that is as close as possible to the average over the one achieved by NEq - indicated with "*". To evaluate the effectiveness of the proposed procedure we focus on the average computational complexity of the back-propagation for a single update iteration (expressed in FLOPs) and the network generalization capabilities at the end of the training. In all the considered scenarios, it is possible to observe a reduction of FLOPs with very marginal or no performance drop for NEq. When compared to the stochastic approach, with fixed probabilities, the amount of saved computation is similar in all the considered scenarios, but the loss in performance varies, depending on the specific architecture/dataset. On the contrary, NEq remains consistent in performance, self-adapting to the specific setup and saving the largest FLOPs for the given performance. Furthermore, testing the stochastic approach with the same FLOPs saving (hence, even letting that information leak in favor of the stochastic approach), the performance loss is lower.

**Limitations.** The current approach analyzes the behavior of an entire neuron. However, empirical experiments show that there could be further improvements considering ensembles of neurons. For example, Fig. 5 shows the average value for the similarities close to a constant but many neurons are still away from the convergence value, meaning that these neurons, at isolation, are still not at equilibrium: is the scenario changing when considering the dynamics of groups of neurons? Furthermore, to validate the adaptability of NEq to the most popular training schemes, no optimization of the hyper-parameters for the training procedure has been performed (as it is out of scope for our evaluation). However, higher savings in computational complexity are possible by tuning the training strategy as well. In such a direction, prospectively, it will be of interest to design more efficient learning strategies which keep into account the concept of neuronal equilibrium.

## 5 Conclusions

The work by Frankle and Carbin (2019) showed the existence of sub-graphs in deep models which, when trained in isolation, can match the original performance of the whole model. Finding these sub-graphs is, however, a complex task, as in the first stages of the learning the model itself is at non-equilibrium. Identifying these with a dynamic strategy, without requiring a posterior over the whole training process, is a crucial task to be solved, towards computational resources saving. Differently from the vast majority of the literature which focuses on the identification of sub-graphs without any concrete computational saving (as they rely on iterative or roll-back algorithms), we have introduced the knowledge of neuronal equilibrium, looking for entire structures of the deep model at equilibrium, not requiring further optimization and gradient computation, which self-adapts to very specific experimental setups on very different learning scenarios. This work opens the doors toward a deeper understanding of the deep neural network's learning dynamics and to the development of new training strategies exploiting this knowledge.

## Acknowledgements

This research was produced within the framework of Energy4Climate Interdisciplinary Center (E4C) of IP Paris and Ecole des Ponts ParisTech. This research was supported by 3rd Programme d'Investissements d'Avenir [ANR-18-EUR-0006-02].

---

[3]`https://github.com/pytorch/vision/tree/main/references/segmentation`

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
