# OpenReview forum: "To update or not to update? Neurons at equilibrium in deep models"
_NeurIPS.cc/2022/Conference — NeurIPS 2022 Accept_

### Official Review · Reviewer_qWfT · 2022-06-27

**Rating:** 7
**Confidence:** 3
**Soundness:** 3 good
**Presentation:** 3 good
**Contribution:** 3 good

**Summary:**

This paper proposes an early stopping scheme for individual neurons in a neural network. The scheme detects neurons at equilibrium by ever-so-often comparing (between successive samples) the dot product of activations on a holdout/validation set, and freezing neurons when this value is zero. (If the value ceases to be zero, the neuron is correspondingly unfrozen.)

The authors choose hyperparameters using an ablation study, and evaluate the main method on four models across three datasets. They show that their method saves a significant amount of computation in exchange for minor reductions in performance.


**Questions:**

It is not clear to me how this algorithm is possible without imposing a great memory overhead – storing the variation of similarities, storing the activation patterns, etc. I would appreciate an accounting of the memory overhead

What is the overall savings in wall-clock time for such computation? In real-world problems, computing dense matrix multiplication is vastly quicker than sparse, and its possible that this reduces the effective savings of the algorithm

What is the standard deviation of the performance and/or savings over a few runs? It is not clear that the differences in performance between approaches (in Table 2) is significant. This is an absolutely critical limitation that must be addressed! (The variation in results is included in Table 1, which is great!)

Neurons may unfreeze during training – what fraction of neurons exhibit this behavior, and how often does this occur?


**Limitations:**

- The significance of the results is not established. This is a critical flaw that should be fixed by presenting a standard deviation over a few runs for each row in Table 2.
- The evaluation is relatively simple, particularly for such a general idea. This should be greatly expanded to better understand the performance of this method.
- It is not clear that the FLOPs savings correspond to real-world savings. This should be tested.

Small points:

 - “Convergency” should be “convergence” (lines 49, 144, 262, others)
 - You can use only the year for the citation in line 64 with “\citeyear”, this avoids repeating the author’s names.




**Strengths And Weaknesses:**

The work seems original, but the idea is close enough to the existing pruning literature that it may have been discovered previously. The paper is written clearly and understandably, with adequate reference to the existing body of knowledge, but the evaluation and discussion should be expanded upon. (See next answers for details.) I’m not certain that the work is significant enough for publication; perhaps it would be better suited to a workshop.

---

> ### Author Response · Authors · 2022-08-01
> **Experiments on multiple seeds, memory consumption, computation for back-propagation, neurons unfreezing**
>
> We would like to thank the reviewer for the pertinent questions and suggestions, thanks to the quality of the work has improved. Below we address the raised concerns individually.
>
> [**Standard deviations for the results**] Thank you for the suggestion. We have added the standard deviations in Table 2, calculated on an average of 5 different seeds. Due to the limited number of characters allowed for a response, the new table can be seen in the revised copy of the paper (where the green elements are the newly added). With the added standard deviations, the experiments still support NEq as a technique to reduce the training resources. Also, the newly obtained average values are still in line with the original values. Please note that the values for CIFAR-10 were already averages over multiple runs, but the standard deviation was omitted. Furthermore, the amount of FLOPs of the baseline and stochastic approach remain unchanged as the number of frozen neurons are defined by the pruning percentage $p$.
>
> [**Memory consumption**] We can estimate the space complexity to store the velocity term for the $i$-th neuron as $\mathcal{O}(\lVert\Xi_{val}\rVert_{0} \cdot N_i)$ where $N_i$ is the size of the output of the $i$-th neuron and $\lVert\Xi_{val}\rVert_0$ is the size of the validation set. As highlighted in Sec. 4.1.3, the validation size can be very small (even lower than one image per class) and we hypothesize that this is possible thanks to the heterogeneity of features extracted from a single image. Indeed, even if we have a single image as input, the sampling points $N_i$ for the equilibrium are a typically large number in convolutional layers. We recommend however to have at least one input image per class in classification tasks, and to have sufficient representation for each class in semantic segmentation tasks. This allows us to only add a trivial memory overhead which does not significantly impact the training. We highlight that this velocity term does not need to be stored in GPU RAM memory and can be stored even in mass memory as it is needed and updated just once per epoch.
>
> [**Computation savings in back-propagation**] In Sec. 2 of the supplementary material (Fig. 1) we provided a benchmark of these times for both a convolutional and linear layer. Please note that our naive implementation lacks the low-level optimization of the CUDA kernels and for this reason we decided to focus on FLOPs to evaluate our procedure in the main paper: however, it is possible in any case to observe a reduction in the computation times depending on the percentage $u$ of the neurons at equilibrium (please note that $u$=100% corresponds to vanilla computation over all the parameters). Please notice also that all the matrices resulting from NEq are *structurally* sparse (entire rows or columns will not be updated); hence the computation for the gradient computation is performed just on a sub-matrix: this gives us the gains reported in the supplementary materials, as there is no computational overhead derived from the sparsity.
>
> [**Neurons unfreezing**] The unfreezing of neurons during training is highly dependent on the training policy employed. For example, in Fig. 3a we can observe how, at around epoch 100, with SGD many neurons freeze, but slowly unfreeze. This is not the case when optimizing with Adam (Fig. 3b), where the freezing is more gradual and smooth, and the neuron’s unfreezing is more rare. While Adam has an adaptive learning rate which self-adjusts depending on the loss landscape, SGD is more subject to stochastic noise.
>
> [**Small points**] We updated the manuscript accordingly, all the updates are marked in green.

---

> > ### Comment · Reviewer_qWfT · 2022-08-03
> > **Thank you for the clarification, and perhaps a link to pruning.**
> >
> > I appreciate the clarification, and the inclusion of reported standard deviation for the results. I also appreciate the clarification that the matrix sparsity is by row or column, which (1) greatly improves the computational efficiency (as most libraries can handle sparsity along one axis easily) and (2) reduces the memory overhead considerably. Thank you for your detailed response and for addressing my concerns! In light of all this, I have increased my rating for your paper to 6.
> >
> > I have a few further comments: first, that the ADAM optimizer includes momentum, and vanilla SGD does not. This is probably the cause of the slow freezing you observe.
> >
> > Second, I skimmed some survey papers in the pruning literature and could not find any prior work that is exactly the same as this. Freezing and reviving neurons has precedent (*Early brain damage.* by Tresp et al., 1997; *Soft Filter Pruning [...]* by He et al., 2018), but (to my knowledge) it has never been combined with your particular decision rule. I encourage you to check through a survey paper in the pruning literature to check that this is true, for example: *What is the state of neural network pruning* (Blalock et al., 2020), and to mention the link to pruning somewhere in your manuscript.
> >
> > And finally, while this should not prevent acceptance of the paper, I also encourage you to perform some small benchmarks (perhaps on one or two architectures) to check that wall-clock time savings match the purported FLOPS savings.

---

> > > ### Author Response · Authors · 2022-08-08
> > > **Link to pruning, ADAM vs SGD and wall-clock time measure**
> > >
> > > Thank you for your kind feedback. It is very well appreciated, and your suggestions really improve the quality of the paper. We address below your additional comments.
> > >
> > > [**ADAM vs SGD**] Very interesting question! In the paper we perform, in Sec.4.1.1, the comparison between ADAM and SGD *with momentum* (as indicated in Sec. 4.1, we have clarified this in the paper). Experimenting vanilla SGD in this section is also an interesting experiment. In particular, in the same setup as in Sec. 4.1.1, we have performed experiments reducing the impact of the momentum $\mu_{opt}$. We have observed that the lower the momentum contribution, the faster the freezing. In particular, already after the first learning rate decay a significant amount of neurons (above 90%) are frozen, while in the first phase almost all the neurons are at non-equilibrium. This happens because the optimizer, in this specific scenario, finds a (sub-optimal) local minimum: indeed the accuracy score drops from 92.96% $\pm$ 0.21% with SGD with $\mu_{opt}=0.9$ to 91.84% $\pm$ 0.09% with vanilla SGD.
> > > We have added the plot visualizing this behavior in the supplementary material (Section 2, Fig. 1).
> > >
> > >
> > > [**Further comparison with pruning literature**] We greatly thank the reviewer for the literature reference provided. We have checked through the works reported in [4] and we confirm the link between NEq and the pruning literature lies in the possibility of “unfreezing” certain parameters. However, there is a major difference between them: while in [1] and [3] the unfreezing corresponds more to “resources re-allocation” to learn new functions (indeed, they are not employed in forward propagation when pruned), in NEq the neuron is kept “in hold” (they are used in forward propagation, but not updated in back-propagation). We have included this additional link to the pruning literature in Section 2 in the “pruning strategies” section and at the bottom of the same section. Below you can find a more detailed commentary on this point.
> > >
> > > The main difference between NEq and other pruning strategies as indicated in [4] lies in their different goal. While pruning imposes a trade-off between model efficiency and performance on the trained task, reducing metrics like memory footprint, storage or computation at inference time or combination of them [4], NEq focuses on the computation reduction at training time, and it can be applied on top of any learning strategy, included any other pruning strategy itself. Despite the difference in the goal, there is a common point: the possibility of “reviving” for parameters/neurons, after being pruned.
> > >
> > > The work by Tresp et al. [1] shows one of the conditions in which the famous *Optimal Brain Damage* by LeCun et al. [2] is sub-optimal (early stopping scenarios), and propose corrections in the “importance scoring function”. In this work the authors suggest the possibility of some pruned parameters to be *revived* in case the minimization of loss deriving from their re-inclusion is sufficiently large. This work focuses the revival (or re-utilization) on single parameters, a concept that is extended to entire neurons/filters by He et al. [3], employing the so-called “soft filter pruning” strategy. Here, while typical pruning approaches set to zero all the pruned filters (and they will never be updated again), with soft filter pruning the removed filters can be re-used for learning, trying to learn completely new functions.
> > >
> > > Both of these works essentially allow re-utilization of pruned weights/neurons to learn *new* functions with the purpose of lowering the loss. On the contrary, NEq freezes the neurons which have learned their input/output function, and unfreezes their update just when there is variation in the input/output relationship, modeled with $\Delta\phi$.
> > >
> > > [**Wall-clock time measure**] We have performed some measurements of the wall-clock time with the popular ResNet-18 trained on ImageNet-1K. We have measured the wall-clock time savings at the back-propagation phase in order to compare it with the theoretical FLOPs saving.
> > > We observe a reduction in the wall-clock time: -17.52% $\pm 0.01%$ (vs the theoretical FLOP reduction -23.08% $\pm 0.08$). We expect that a more optimized implementation of the layers (for example at CUDA kernel level) will lead to a greater time reduction, closer to the theoretical one. We have included this measure in the supplementary material.
> > >
> > > **References**
> > >
> > > [1] Tresp, Volker, Ralph Neuneier, and Hans-Georg Zimmermann. "Early brain damage." Advances in neural information processing systems 9 (1996).
> > >
> > > [2] LeCun, Yann, John Denker, and Sara Solla. "Optimal brain damage." Advances in neural information processing systems 2 (1989).
> > >
> > > [3] He, Yang, et al. "Soft filter pruning for accelerating deep convolutional neural networks." (2018).
> > >
> > > [4] Blalock, Davis, et al. "What is the state of neural network pruning?." Proceedings of machine learning and systems 2 (2020): 129-146.

---

### Official Review · Reviewer_MVNQ · 2022-07-03

**Rating:** 3
**Confidence:** 3
**Soundness:** 3 good
**Presentation:** 3 good
**Contribution:** 2 fair

**Summary:**

The authors propose a method to selectively choose which neuron's weights to update during training, based on the concept of Neuron Equilibrium.
The main idea is to stop backpropagation through neurons that reach equilibrium, so that no more (possibly useless, but compute-wasteful) backpropagation operations are performed.
While the idea is intriguining, I would imagine the effect to have minimal impact on actual training times (i.e., no more than 1/2-1/3x the original training time, which may not be worth the implementation overhead; see below in Weaknesses); it would be very useful to see a comparison of wallclock training times.


**Questions:**

See 'weaknesses' above: what is the difference in wallclock training time between a baseline network trained in a typical way, vs using your method?

**Limitations:**

See 'weaknesses' above.

**Strengths And Weaknesses:**

Strengths:
- The idea of working at a neuron instead of connection level, while not per se novel, is useful and can indeed be beneficial.

Weaknesses:
- While training smaller networks or pruning them during training can improve training time of neural networks, it often is not crucial to do. As far as I am aware, network pruning is mostly useful to obtain smaller models for cheaper inference, or for inference on less powerful devices. In those cases, pruning can be just performed after training the larger network.
- Following on my previous point: the main benefit of the authors' approach is that fewer operations are required during training, as the disabled neurons are only used in feedforward activation and not in backpropagation (which usually take a few times more operations than the corresponding feedforward ones). It would be useful to see a comparison of wallclock training times rather than just Backpropagation FLOPS. Looking at wallclock times should be especially interesting since the induced sparsity in backpropagation may actually lead to computational overheads on modern GPUs, where large matrix multiplications are very fast.

Minor:
- Watch out some language issues, e.g., "like the same model already at convergency" -> "at convergence".

---

> ### Author Response · Authors · 2022-08-01
> **On the benefit of NEq and FLOPs instead of wall clock time**
>
> We thank the reviewer for the provided commentary, below we answer the posed questions.
>
> **[Improving training time is often non crucial]** With modern models becoming bigger and bigger, reducing the complexity of the training process is beneficial in order to save resources, and also shows an environmental impact [3]. In recent years, techniques with the aim of reducing the training complexity have been defined  [1, 2], but the achieved results are not yet satisfactory.
>
> In our work we explore (and exploit) the oscillatory behavior of the model's neurons during training, which is a *novel* finding and opens the road towards new research directions related to training optimization of neural networks. Understanding that, after certain learning phases, a part of the model has already learned its input-output function while other parts are still at non-equilibrium, is a step towards a deeper understanding of the dynamics of neural networks training.
>
> Concerning the prevalence of post-training pruning, we believe that reducing the training’s resource requirements and pruning are not mutually-exclusive (and, on the contrary, they are jointly exploited, like in [4]) and applying both procedures can be beneficial during the whole life-cycle of the network.
>
> **[Wall clock training times instead of FLOPs]** In Sec. 2 of the supplementary (Fig. 1), we provide an empirical evaluation for convolutional and linear layers in which we show that, disabling the backpropagation for a percentage of neurons, leads to a speedup. In the figure, $u=100$% represents the standard vanilla training and all the results are averaged over 1000 measurements. It is worth noting that our implementations are not on par with low-level CUDA kernels, optimized for matrix operations, and for this reason we decided to focus on the more implementation-agnostic FLOPs when evaluating the models; hence, we expect even higher performance with a lower level implementation. Also, please note that the matrices resulting from the application of NEq are matrices in which entire rows (or columns) are removed. Hence, the update happens on a (dense) sub-matrix, and this does not lead to any particular computational overhead. This is the motivation behind the gain in computation evaluated.
>
> **[Language]** We corrected the errors, the changes are highlighted in green in the revised submission.
>
> [1] Utku, Evci, et al. "The Difficulty of Training Sparse Neural Networks". ICML 2019.
>
> [2] Jonathan Frankle, et al. "Pruning Neural Networks at Initialization: Why Are We Missing the Mark?." International Conference on Learning Representations. 2021.
>
> [3] Strubell, Emma et al. "Energy and Policy Considerations for Deep Learning in NLP." Proceedings of the 57th Annual Meeting of the Association for Computational Linguistics. Association for Computational Linguistics, 2019.
>
> [4] Shen, Maying, et al. "When to Prune? A Policy towards Early Structural Pruning." Proceedings of the IEEE/CVF Conference on Computer Vision and Pattern Recognition. 2022.

---

### Official Review · Reviewer_wKon · 2022-07-09

**Rating:** 6
**Confidence:** 3
**Soundness:** 3 good
**Presentation:** 3 good
**Contribution:** 3 good

**Summary:**

The authors investigate a question that fits within into a recent fascinating line of investigation - namely that it is possible to find subsets of neurons with a neural network that, when trained, match the performance of the full network. Being able to rapidly identify this subset of neurons early on in training would enable networks to be trained at a much lower cost. Unfortunately, so far it has been challenging to identify this subset until late in training, at which point it no longer helps.

The authors pursue a related line of enquiry - is it possible to instead determine that, at a given point in training, a given neuron is "at equilibrium"? This is to say that its activity when driven by a specific input is consistent between training updates. The hypothesis is that such neurons can be excluded from training updates until they exit equilibrium.

The authors investigate this question empirically, and find some promising results suggesting that this is the case. They test their methods for several different architectures and tasks, finding consistent performance improvements in terms of FLOPS over the baseline. They compare this to a simpler approach in which subsets of neurons are randomly dropped out of training at each epoch. The improvement over this stronger baseline is less significant, although still measurable.

**Questions:**

My most important question is the following - what precisely is t in e.g. Eqn 1? Is this before and after training for a given epoch? If so, I don't understand the limit of t->inf in eqn. 2. Relatedly, there are not enough details on the precise algorithm (an algorithm box would be appreciated). In order to be able to decide to "reenable" neurons that have come out of equilibrium, presumably you need to do steps of gradient descent with all neurons being updated? All of these details were unclear to me.

Just below eqn 2, three different conditions on k are laid out. What is the importance of distinguishing between [0,1] and [-1, 0]? This wasn't made particularly clear (I also assume a typo -> (k \in (0;1) should be k \in [0;1]).

Its not clear that all the additional machinery of eqns 3 to 5 is necessary. E.g. Fig 4 suggests that \mu=0 works more or less just as well.

I might naively have assumed that equilibrium would mean k=1. This seems to be the case for essentially no neurons until epoch 150. Do you have a sense of why so many neurons are apparently oscillatory?

"We can extend this dimensioning for the momentum coefficient to any layer assuming that the neurons in the trained model will reach at most equilibrium with de-correlated outputs." - this sentence wasn't super clear to me.

Table 1 is missing a caption, and in particular it would be useful to clarify that FLOPs is averaged across training.

The title of section 4.2 (experimental setup) is a bit confusing, given that you have just talked about an experiment in the preceding sections.

In the limitations section, its a bit unclear how you would extend your line of reasoning to consider groups of neurons.

**Limitations:**

As mentioned above, the proposed method only reduces training FLOPs. I further don't have a good sense of how significant the demonstrated savings are as compared to other SOTA techniques.

I don't know of any potential negative societal impacts from this work.

**Strengths And Weaknesses:**

I am not an expert in the field, but I found the core results interesting, seemingly novel, and empirically robust. The methods they suggest seem fairly straightforward to implement.

In terms of weaknesses, there are several (crucial) aspects of the paper that are unclear -- these are outlined below. In addition, the proposed methods only reduce the FLOPs overhead for training, but do not reduce the network inference cost. This is in contrast to methods that find sparse subnetworks. Finally, I would have liked to see some more discussion on the practical implications of such techniques. As a non-expert, I might naively assume that these techniques would not necessarily lead to significant performance improvements on specialist accelerators. I suspect this is the case for many related techniques, but it would be nice to have these potential limitations more explicitly laid out.

---

> ### Author Response · Authors · 2022-08-01
> **Correlation vs decorrelation, the velocity term, many neurons at non-equilibrium**
>
> We would like to thank the reviewer for his comments, thanks to the readability of the paper has improved. Here follow the answers to the reviewer's questions.
>
>
> [**Meaning of index $t$ and algorithm**] $t$ is a temporal index, more specifically for this work, an epoch index. The limit in (2) should be interpreted as a limit for a sufficiently large number of epochs (or iterations in a more general way). For the algorithmic structure, please refer to Fig.2. In particular, the steps to be followed are:
> ```
> Train with equilibrium detection():
> - for t in epochs:
> - - train the model;
> - - if end of the training:#eg. wall epoch for training or other condition
> - - - return;
> - - compute equilibrium for neurons (4)#using a validation set, updating velocity
> - - threshold and evaluate which neuron is at equilibrium (6)#thresholding over a $\varepsilon$
> ```
>
> We will include this pseudo-code in the supplementary material.
>
> [**Difference between correlation, decorrelation and anticorrelation in (2)**] Interpreting the output of a neuron in a vectorial form, we can identify regimes of maximum correlation ($k=1$), decorrelation ($k=0$) and anti-correlation ($k=-1$). Of these, the two most important ones are maximum correlation (meaning that the output of the neuron remains fixed) and decorrelation (meaning that it changes completely). Having maximum anti-correlation means essentially having exactly the opposite feature extracted; which for instance has impacts on the way the model itself processes the information (example, with sign swaps), but not on the pure information. Indeed, this distinction is in most of the cases not even possible (like in ReLU networks, where $\phi_i \in [0; 1]$) and in our final formulation as in (4) it is absorbed by analyzing the variation of $\phi_i$ (3).
>
> [**The importance of the velocity term (with $\mu_{eq}\neq 0$)**] The use of a velocity term as in (4) smoothens the estimation over the equilibrium in (1). In such a way, we help in discouraging noisy estimates over the equilibrium as in Fig.4 (left) epochs 150-155. Indeed, in cases where the learning rate policy is non constant but decayed, without the velocity term the estimation over the neurons at equilibrium is noisy and sub-optimal (from the epoch 160 onward, the orange line (with $\mu_{eq}=0.5$) is below the blue one (without velocity), meaning that the pruned FLOPs are lower with momentum).
>
> [**So many neurons at non-equilibrium**] The neurons are considered at equilibrium when the variations over $\phi^t$ are constant, meaning that $\Delta \phi^t = \phi^t - \phi^{t-1} = 0$ (3). The question over the oscillatoriety of the neuron is very interesting, and some insights are visible from Sec.4.1.1, where on the same conditions the optimizer is simply changed (SGD-Adam). In particular, we observe in Fig.3 that with SGD a large number of neurons remain at non-equilibrium even after the first learning rate decay step (epoch 100), while the neurons reach equilibrium progressively with Adam. This is explained by the nature of the two optimizers themselves, where Adam scales the update step for every parameter depending on the individual gradient, while SGD (with momentum) is more subject to stochastic noise. The latter is particularly evident after the first learning rate decay (epoch 100), where some neurons at equilibrium, starting from epoch 110, become at non-equilibrium (green curve Fig. 3a with increasing trend) because of the stochastic noise induced by the optimizer.
>
> [**Dimensioning of $\mu_{eq}$ in non ReLU-activated models**] The dimensioning for $\mu_{eq}\in [0, 0.5]$ is valid for ReLU-activated networks, where the output can be positive or, at least, zero. In such a case, any scalar product between $\boldsymbol{y_i^{t}}$ and $\boldsymbol{y_i^{t-1}}$ can be in range [0; 1] only. According to our definitions in Sec. 3.1, the output will be at least decorrelated, as it is impossible to have anticorrelation. When dealing with activations which can be also negative values (and for instance, anticorrelation is also allowed), the momentum coefficient should be in range  $\mu_{eq}\in [0, 0.25]$.
>
> [**Caption in Table 1**] The caption is unique for both the sub-tables. We have added sub-captions indicating the single sub-tables independently.
>
> [**Title of Sec.4.2**] We have modified the title in “Main experiments”. Thank you for for raising the issue.
>
> [**Addressing equilibrium to groups of neurons**]This work has observed that many neurons can reach an equilibrium state as defined in (2). This formulation is limited in the analysis of one single neuron at a time (the variation of $\phi_i$ is observed over multiple samples, but independently at the single neuron scale). Towards this end, high-order interactions between neurons (like for example, two neurons might “swap” in their behavior) are not taken into account. Implementing this analysis in an efficient way is an interesting research question and a challenge for next work.

---

### Official Review · Reviewer_2pYr · 2022-07-09

**Rating:** 6
**Confidence:** 4
**Soundness:** 4 excellent
**Presentation:** 3 good
**Contribution:** 3 good

**Summary:**

This paper decreased the trainable neurons in each epoch by exploiting a concept of neuronal equilibrium. If the estimated velocity of the similarity variation between the output of a neuron at time $t$ and $t-1$ is smaller than a threshold, the neuron is regarded as in the equilibrium state and will stop updating its own parameters. After testing the proposed method on different state-of-art tasks, results show that the proposed method can significantly decrease the FLOPs during back propagation with marginal or no performance drop, which is much better than randomly stopping the update of a subset of neurons.

**Questions:**

1. Evaluating the equilibrium state for neurons also requires additional computational cost. Could you estimate the time complexity of the equilibrium evaluation + the back propagation using NEq and compare it with the time complexity of the baseline back propagation method? Then readers may understand how much it will decrease overall computational cost.
2.  If the momentum coefficient $\mu_{eq}$ is not zero, the similarity will still vary even if $v^t = 0 \rightarrow \Delta \phi^t = \mu_{eq} v^{t-1}$. Why can this state be regarded as equilibrium?
3. The number of validation images is different in different experiments. Does the increase of the validation image affect the time cost severely? How do we choose a reasonable validation set size during training?


**Strengths And Weaknesses:**

Strengths:
1. The proposed method can decrease the training FLOPs significantly with very marginal or no performance drop.
2. Compared to traditional pruning method, this method doesn't require additional fine-tuning process and thus simplifies the training process.
3. The proposed method is very straightforward and easy to apply to different methods.

Weaknesses:
1. Evaluating the equilibrium state for neurons also requires additional computational cost. The time complexity of the equilibrium evaluation + the back propagation using NEq is unclear yet.
2. The theoretical basis of the momentum coefficient is unclear.
3. The effect of the number of validation images is unclear.

---

> ### Author Response · Authors · 2022-08-01
> **On NEq's complexity, equilibrium states and validation set size**
>
> We would like to thank the reviewer for the very insightful questions, which we address below.
>
> 1- The evaluation of the time complexity in the computation of the equilibrium (for the given $i$-th neuron) depends on two factors: the size of the output $N_i$ and the number of samples in the validation set $\lVert\Xi_{val}\rVert_0$. For the computation of $\phi_i$, the complexity will be $\mathcal{O}(\lVert\Xi_{val}\rVert_{0} \cdot N_i)$. For the variation of $\Delta \phi_i$, as it is a simple subtraction, the complexity is simply $\mathcal{O}(1)$ as well as the subtraction with the velocity term in (4). We conclude that the time complexity for the equilibrium evaluation for every neuron is $\mathcal{O}(\lVert\Xi_{val}\rVert_{0} \cdot N_i)$ (which is easily parallelizable on GPU).
>
> For back-propagation part, the complexity is actually reduced, as the gradient computation for the neurons at equilibrium does not have place. In particular, let us define $M_i$ as the input size for the $i$-th neuron. The (local) back-propagation complexity for the $i$-th neuron (from its output $\boldsymbol{y_i}$ to its parameters $\boldsymbol{w_i}$) is $\mathcal{O}(\lVert\Xi\rVert_{0} \cdot N_i \cdot M_i \cdot \lVert \boldsymbol{w_i}\rVert_{0} \cdot u_i)$, where $u_i$ is a binary variable indicating whether the $i$-th neuron is being updated ($u_i=1$) or whether it is at equilibrium ($u_i=0$) and $\lVert\Xi\rVert_{0}$ is the batchsize. Intuitively, the higher the percentage of neurons at equilibrium, the lower the complexity for back-propagation. A benchmark for the times of the back-propagation on fully-connected and convolutional layers is reported in Section 2 of the supplementary material (Fig. 1 of the supplementary material). In the Figure, $u$=100% corresponds to the vanilla gradient computation, and all the results are averaged over 1000 measurements.
>
> ~
>
> 2- We absolutely agree with the interpretation of the reviewer. Let us analyze the effect of such a condition.
>
> Given $v^t=0$, in the timestep $t+1$ we will have
>
> $v^{t+1}=\Delta \phi^{t+1}$ (*)
>
> For instance, unless $\Delta \phi^{t+1}=0, v^{t+1}\neq 0$, meaning that there is no variation in the value of $\phi_i$, the neuron comes back alive as it is at non-equilibrium.
>
> Focusing on the timestep $t$, we have
>
> $\Delta\phi^t=\mu_{eq}\cdot v^{t-1}$ (**)
>
> The obvious solution is here to have $\Delta\phi^t = 0$, which corresponds to our equilibrium definition. Let us investigate a solution such that $\Delta\phi^t \neq 0$: we can rewrite (**) as
>
> $\Delta\phi^t=\sum_{i=1}^{\infty} (-1)^{i-1}\Delta\phi^{t-i}\mu_{eq}^i$ (***)
>
> This happens when the variation at time $t$ is comparable with the weighted average variation in the previous time-steps, suggesting an oscillatory state also in such a case. According to (*), in case the reached equilibrium is not maintained (as $\Delta \phi^{t+1}\neq 0$), then the neuron will be updated again.
>
> ~
>
> 3- The space complexity to store the velocity term for each neuron is $\mathcal{O}(\lVert\Xi_{val}\rVert_{0} \cdot N_i)$, but as highlighted in Sec. 4.1.3,  $\lVert\Xi_{val}\rVert_{0}$ can be extremely small, and provides comparably good results even with just size one. We hypothesize that this is possible thanks to the heterogeneity of features extracted from a single image: even if we have a single image as input, the sampling points $N_i$ for the equilibrium are a typically large number in convolutional layers. We recommend however to have one input image per class in classification tasks, and to have sufficient representation for each class in semantic segmentation tasks.

---

> > ### Comment · Reviewer_2pYr · 2022-08-08
> > **Increase score to 6**
> >
> > I appreciate authors' detailed responses. I am happy to increase the score to 6.

---

### Meta-Review · Area_Chair_VP6X · 2022-08-26

**Recommendation:** Accept
**Confidence:** Less certain

**Metareview:**

Despite there not being complete agreement on the novelty of the method presented in the paper, most Reviewers praised the idea of proposing an early stopping scheme that is based on reusing concepts from the pruning literature but with the innovation of shifting the focus from connections to units.
A major criticism moved in the initial reviews was due to doubts on the contribution of the work and in particular its practical benefit in terms of training speedup. However, the rebuttals did a good job at dispelling these doubts.
Following the rebuttals, the paper garnered unanimously positive scores among Reviewers, owing to its significance for bridging the pruning literature with practical training speedup supported by convincing empirical evidence.

**Award:**

No

---

### Decision · Program_Chairs · 2022-09-14

Accept